# Effect of Home-Based Training with a Daily Calendar on Preventing Frailty in Community-Dwelling Older People during the COVID-19 Pandemic

**DOI:** 10.3390/ijerph192114205

**Published:** 2022-10-30

**Authors:** Misa Nakamura, Masataka Ohki, Riku Mizukoshi, Itsuki Takeno, Taira Tsujita, Ryota Imai, Masakazu Imaoka, Masatoshi Takeda

**Affiliations:** 1Graduate School of Rehabilitation, Osaka Kawasaki Rehabilitation University, Kaizuka, Osaka 597-0104, Japan; 2Department of Rehabilitation, Osaka Kawasaki Rehabilitation University, Kaizuka, Osaka 597-0104, Japan

**Keywords:** COVID-19, frailty, calendar, community-dwelling, older people

## Abstract

It has been reported that marked decreases in physical activity including social activities, deterioration in eating habits and mental health, and an increase in frailty have occurred during the COVID-19 pandemic. This study aimed to devise a method to prevent the onset and progression of frailty during the COVID-19 pandemic and to verify its effect. The subjects were 111 community-dwelling older people who answered questionnaires before and after the intervention. A calendar incorporating 31 different tasks, one for each day, was created as an intervention tool with the aim of improving motor, oral, and cognitive functions. The intervention group (*n* = 49) participants performed these tasks every day for 3 months. The primary outcome was the Kihon checklist (KCL) score. When the amount of change in the KCL score before and after 3 months was compared between the two groups, no difference in the total score was observed between the two groups; however, the intervention group showed significantly improved cognitive function in the KCL sub-domain. In the intervention group, the number of pre-frailty and frailty patients decreased significantly after the intervention compared to before the intervention. These results suggest that the use of the calendar created in this study during the COVID-19 pandemic could prevent decreased cognitive function in the KCL sub-domain and could help prevent the onset and progression of pre-frailty and frailty.

## 1. Introduction

COVID-19 has spread worldwide. Even today, Japan faces daily fluctuations in the number of people who are infected. In July 2022, the World Health Organization (WHO) reported that Japan had the highest number of COVID-19 infections per week in the world, at approximately 970,000 [1].

Frailty is defined as a state of enhanced vulnerability to external stress due to various organ dysfunctions associated with aging. Frailty includes not only physical problems such as muscle weakness and malnutrition in old age but also mental and psychological problems such as cognitive impairment and depression, as well as social problems such as living alone or being confined to one’s home, and economic hardship, future falls, impairment of daily living functions, and hospitalization. It is a pathological condition that leads to various poor outcomes such as decreased life expectancy [2]. It has also been reported that, with appropriate interventions, frail persons can be returned to robust [3,4,5].

Beginning with the restrictions on going out under the declaration of a state of emergency issued in April 2020, self-restraint of social activities has continued. In such circumstances, many reports of mental and physical changes of community-dwelling older people have been published. Prominent decreases in physical activity and increased frailty have been reported [6,7,8,9]. Yamada et al. reported that socially inactive older adults living alone are more likely to experience incident frailty/disability due to reduced physical activity during the COVID-19 pandemic [8]. In addition, cognitive function of the Korean elderly cohort declined much more during the pandemic than before the pandemic, particularly in terms of memory and recall function [10].

At present, since the end of the COVID-19 pandemic is unpredictable, it is extremely important for all people to maintain their health while balancing infection prevention and activities to ensure the future health of society. For that purpose, various optimal health promotion methods need to be developed and promoted.

Therefore, this study aimed to devise a method to prevent the onset and progression of frailty, which can be induced by the special social environment of the COVID-19 pandemic, and to verify its efficacy in community-dwelling older people in Japan.

## 2. Materials and Methods

### 2.1. Participants

The subjects were local older people who participate in physical function measurement meetings in Kaizuka City every year [11]. Participants answered the questionnaires in July 2021 and November 2021. The preliminary questionnaires were sent to 250 people in July, and the 126 people who returned the questionnaires were divided into a control group (*n* = 64) and an intervention group (*n* = 62). The intervention group completed a calendar task for 3 months. In November, the post-event questionnaire was sent out. The analysis included 62 people in the control group and 49 people in the intervention group who responded to the survey. The data of these 111 people were used in the final analysis (Figure 1). Referring to previous studies [12], the sample size was calculated post hoc using Gpower version 3.1.9.7 computer software with a two-tailed significance level of 0.05, and a power of 0.8; the estimated number of people was 90, and the 111 people included in this study were thus considered appropriate. Participants were allocated using a stratified randomization method, with stratification by sex, age and frailty level.

This study was approved by the Ethics Committee of Osaka Kawasaki Rehabilitation University (Approval No. OKRU-RB0002) and was conducted in accordance with the Declaration of Helsinki. This study was also registered with the UMIN Clinical Trial Registry System for intervention research (ID: UMIN000045102).

### 2.2. Intervention

A specialized daily calendar was sent to the intervention group, and they performed the tasks in it for 3 months (Figure 2). The calendar incorporated 31 different tasks for improving motor, oral, and cognitive functions. When developing this calendar, the aims were (1) to incorporate content that promotes motor functions (squats, towel exercises, etc.), oral functions (cheek exercises, tongue exercises, etc.), and cognitive functions (word rearranging, finding word mistakes, etc.); and (2) the content should be simple so that many older people can do it alone at home. Finally, one month consisted of 21 days of motor function, 6 days of oral function, and 4 days of cognitive function tasks (Table 1). The proverbs and their explanations were in the daily calendar. The intervention group used this one-month calendar three times in the 3-month period whereas the control group continued their normal life without using the calendar for 3 months.

### 2.3. Outcome

Outcomes were scored on the Kihon checklist (KCL) [13]. The KCL is a self-administered questionnaire in which participants answer “yes” or “no” to 25 questions about living conditions and physical and mental functions. It consisted of a 5-item assessment of activities related to daily life (“life function”), a 5-item assessment of locomotor function (“physical function”), a 2-item assessment of malnutrition (“nutrition”), a 3-item assessment of oral function (“oral function”), a 2-item assessment of outdoor activities (“outdoor activities”), a 3-item assessment of cognitive function (“cognitive function”), and a 5-item assessment of depressive mood (“depression”), with a set of 7-domain questions. In the content of each question, 1 point was added when it was considered that there was a problem with each item, and the higher the score, the more likely there was a problem with life function. Recently, the KCL has been used internationally as a frailty assessment; when the patient’s total KCL score is 0–3 points, the patient is diagnosed as robust, 4–7 points as prefrailty, and 8–25 as frailty. Receiver operating characteristic curve analyses showed that the areas under the curves for the evaluation of frailty status were 0.81 (sensitivity, 70.3%; specificity, 78.3%) for prefrailty and 0.92 (sensitivity, 89.5%; specificity, 80.7%) for frailty at total KCL scores of 3/4 and 7/8, respectively [14].

### 2.4. Statistical Analysis

Statistical analysis was performed using Student’s *t*-test and Pearson’s chi-squared test for differences between the control and intervention groups. Two-way repeated analysis of variance (ANOVA) was carried out to compare changes in KCL total score and sub-domain score values to determine the effect of intervention. Effect sizes were indicated by partial eta-squared (η^2^). In addition, the McNemar-Bowker test was also performed to clarify the difference in the number of robust and pre-frail-frail populations for 3 months. SPSS Statistics software (version 26; IBM Corp., Armonk, NY, USA) was used as the statistical analysis software.

## 3. Results

### 3.1. Characteristics of the Study Participants

The characteristics of the participants in the control group and the intervention group are shown in Table 2. A total KCL score of three points or less was considered robust, and a total score of four points or more was considered pre-frailty and frailty (pre-frail-frail). There were no significant differences in the sex ratio, age, solitary living rate, and number of pre-frail-frail patients between the two groups (Table 2). In the intervention group, 23 people (47.0%) used the calendar every day, 9 people (18.4%) used it 5–6 days a week, 8 people (16.3%) used it 3–4 days a week, and 9 people (18.4%) used it 1–2 days a week. All people used the calendar during the 3 months, with daily use having the highest rate.

### 3.2. Comparison of KCL Total Score and Sub-Domain Score Changes in the Two Groups

Table 3 shows the comparison of KCL total score and sub-domain score changes in the two groups. There was no difference in the amount of change in the total KCL score before and after 3 months between the two groups, but “cognitive function” showed a significant improvement in the intervention group (F = 4.347, *p* = 0.039, partial η^2^ = 0.038) (Table 3).

### 3.3. Comparison of the Number of Participants Who Were Robust and Pre-Frail-Frail Pre-Intervention and Post-Intervention

The numbers of robust and pre-frail-frail participants before and after 3 months in each group were compared. There was a significant difference in their ratio in the intervention group (*p* = 0.035) (Table 4). In other words, the number of pre-frail and frail participants was less after the intervention than before the intervention.

## 4. Discussion

In the present study, the intervention group showed significantly improved cognitive function in the KCL sub-domain. Furthermore, the number of pre-frailty and frailty patients decreased significantly after the intervention compared to before the intervention.

The cognitive function domain consists of question items related to “memory”, “executive function”, and “date orientation”, and it is reportedly useful as a mild cognitive impairment screening tool [15]. In addition, examining the relationship between this domain and the onset of dementia, it has been reported that the higher score of this domain, the higher the risk of developing dementia [16]. The present study showed that cognitive function deteriorated in the control group and improved in the intervention group. This result is considered to be due to reading and thinking about proverbs written in the calendar every day in addition to the effect of training to improve cognitive function during the task.

There have been several reports of interventions for frailty prevention during the COVID-19 pandemic. A health class run by residents who exercise while watching a video recording of simple gymnastics can be regarded as a place for interaction with people for a year during the COVID-19 pandemic, and improvements in oral function, outdoor activities, cognitive function, and a depressed mood in KCL domains have been reported [17]. Reports related to physical frailty showed that home exercise programs improved motor function and lower extremity muscle strength [18,19], and TV-based assistive integration technology improved physical and mental well-being [20]. Peretz et al. reported that maintaining social networks and reading contributes to maintaining physical activity [21]. In addition, the National Center for Geriatrics and Gerontology in Japan expects to determine appropriate activity plans to prevent physical and mental decline at home for older persons who cannot go out or have limited social activities during the COVID-19 pandemic [22]. In the present study, the number of pre-frail-frail participants after the intervention decreased significantly compared to before the intervention in the intervention group, which strongly supports the results of the before and after changes in the KCL score described above. Based on the results of the KCL domain mentioned above, this intervention’s effect may have been due to psychological factors including cognitive function rather than physical factors.

In community-dwelling older adults, regardless of age, sex, polypharmacy, undernutrition risk, and frailty status, information and communication technology (ICT) users were more proactive in maintaining their health during the COVID-19 pandemic [23]. However, in our previous survey targeting older people in the study region, it was found that the ownership rate of ICT devices such as tablets (7.6%) and personal computers (20.9%) was very low [24]; therefore, a daily calendar was created. The point in creating a daily calendar was to include tasks that promote motor function [25], oral function [26], and cognitive function [27], which are closely related to frailty. The use of the calendar was effective in improving cognitive function, but it was not effective in improving oral function or motor function. In the future, it will be necessary to consider the content of calendar assignments and the period to be used.

The response rate for the post hoc questionnaire was 79.0% in the intervention group and 96.9% in the control group. The low response rate in the intervention group may be due to non-response of those who did not use the calendar during the period. The percentage of pre-frail-frail participants in the present study was 69.4% at baseline. In the previous report, the results for women were slightly higher than 60.3% [18], which may be partly because the baseline period of the present study was during the ongoing COVID-19 pandemic.

It has been proposed that it is important to address frailty prevention from the three aspects of nutrition, exercise, and social activity during the COVID-19 pandemic [28]. Although the task calendar created in the present study is an excellent method for being able to do it alone at home, it impedes social involvement. It would be most effective to incorporate the various intervention methods described above, including the present method, according to the risk of infection with social activities and the degree of movement restriction during the COVID-19 pandemic.

This study has a major limitation. Because the KCL was self-administered, there were some missing values in the returned responses, so the participants were asked about the missing values by telephone. Therefore, some responses were not necessarily made in the same environment, which may decrease the reliability of the data. However, this is an unavoidable problem in this type of research. There is also a lack of research on pre- intervention and post-intervention nutritional status, the socio-family situation of individuals, and the degree of dependency. In addition, as mentioned above, it is possible that the analysis results were only for those who were highly receptive to this intervention method. Therefore, in the future it will be necessary to conduct an evaluation that also considers receptivity. According to the trend in the number of people infected with coronavirus, a peak increase in the number of infected people called the 5th wave was seen from early August to mid-September in 2021, and a state of emergency was declared from 2nd August to the end of September. After that, there were no particularly strict restrictions on going out until the request for measures to prevent the spread of the virus in late February 2022 [29]. In the future, it will be necessary to consider intervention methods based on such differences in social situations.

## 5. Conclusions

During this research, social conditions strongly influenced people’s lifestyles and behaviors, so it is unwise to draw a general conclusion. However, performing the task created for frailty prevention every day during the COVID-19 pandemic is expected to prevent deterioration in cognitive function in the KCL sub-domain and to help prevent the onset and progression of pre-frailty and frailty.

## Figures and Tables

**Figure 1 ijerph-19-14205-f001:**
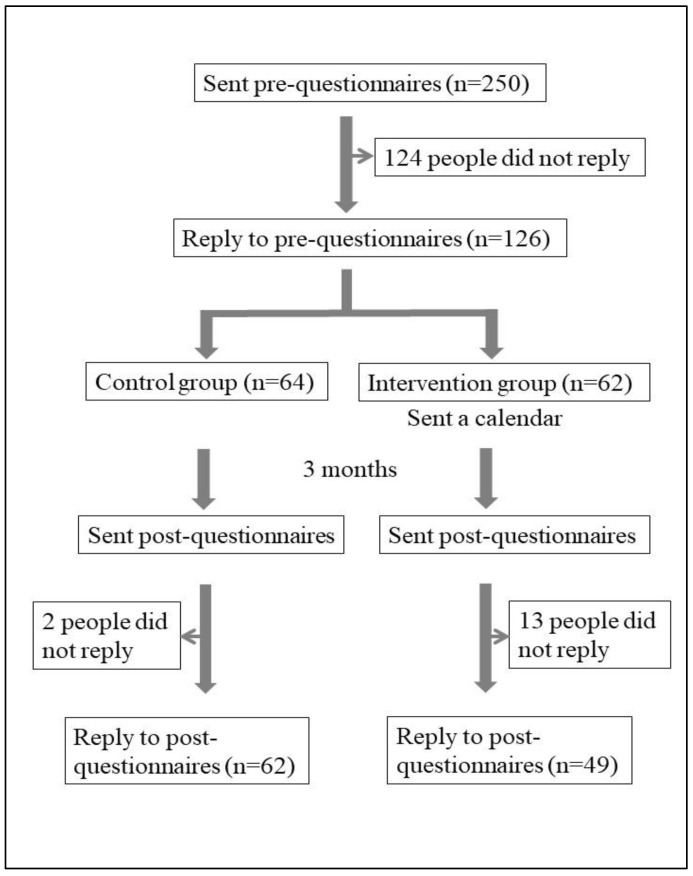
Flow chart showing the participants recruited in the study.

**Figure 2 ijerph-19-14205-f002:**
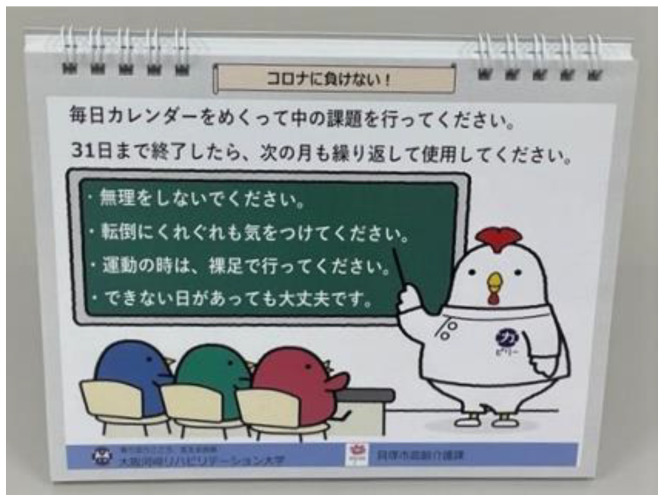
The calendar that was created. Notes on doing calendar assignments are described. Don’t overdo it. Be careful not to fall. Exercise barefoot. Don’t worry if you can’t complete the task.

**Table 1 ijerph-19-14205-t001:** Daily assignments for a month in the calendar.

Day	Category	Contents and Times ^‡^
1	Physical	Standing on one leg ([1 min × 3 times/one leg] × 2)
2	Oral	Cheek exercises (Puff up and purse × 10 times)
3	Physical	Toe-up (10 times)
4	Physical	Heel-up (5 times)
5	Physical	Back muscle training (5 s × 5 times)
6	Oral	Tongue exercise-1 (put out or retract 10 times)
7	Physical	One knee extension ([3 s × 5 times/one knee] × 2)
8	Cognition	Color reading (1 time)
9	Physical	Opening and closing movements of the hip joint (10 times)
10	Physical	Inner thigh movement (3 s × 5 times)
11	Oral	Tongue exercise-2 (move the tongue up and down, left and right × 10 times)
12	Physical	Side leg lift ([5 s × 10 times/one leg] × 2)
13	Physical	Stretching with a towel (5 times × 2)
14	Cognition	Sorting characters (1 time)
15	Physical	Back thigh stretches (3 s × 5 times)
16	Physical	Lunge exercise (10 times/one leg × 2)
17	Physical	Strength training using stairs (3 min)
18	Cognition	Find word differences (1 time)
19	Physical	Abdominal muscle exercise (5 times)
20	Physical	Squat (5 times)
21	Oral	Tongue exercise-3 (turn the tongue 10 times)
22	Physical	Calf stretch (5 times/one leg × 2)
23	Physical	Buttocks up (5 s × 5 times)
24	Physical	Getting up from a chair (5 times)
25	Cognition	Calculation (5 questions)
26	Physical	Step backwards ([5 s × 5 times/one leg] × 2)
27	Oral	Mouth muscle training-1 (Pronounce “Pa”, “Ta”, “Ka”, and “Ra” 10 times each)
28	Physical	Shoulder and back muscle training (5 times forward and backward)
29	Physical	Thigh lift (3 s × 5 times)
30	Oral	Mouth muscle training-2 (Pronounce “A”, “I”, “U”, “E”, and “O”, and tongue twister)
31	Physical	Core training (3 times/one leg × 2)

^‡^ Recommended to do it at morning, noon and night.

**Table 2 ijerph-19-14205-t002:** Characteristics of the study participants at baseline.

	Control		Intervention		*p*	Effect Size
Participants (n/male%)	62	27.42	49	28.80		
Age (y, mean/SD)	76.27	5.12	77.61	6.03	0.209 ^†^	0.130 (r)
Living alone (n/%)	12	19.36	13	26.53	0.909 ^§^	0.095 (φ)
Frailty status						
Robust (n/%)	20	32.26	14	28.57	0.676 ^§^	0.040 (φ)
Pre-frail-frail (n/%)	42	67.74	35	71.43		

Pre-frail-frail, pre-frail, and frail. Comparing each value between control and intervention by the ^†^ Student’s *t*-test or ^§^ Pearson’s chi-squared test.

**Table 3 ijerph-19-14205-t003:** Comparison of KCL total score and sub-domain score changes in the two groups.

	Control		Intervention		Two-Way Repeated ANOVA			
	Pre	Post	Pre	Post		F	*p*	Effect Size (Partial η^2^)
KCL total Mean (SD)	5.39 (3.65)	5.13 (3.55)	5.31 (2.94)	4.37 (2.82)	interaction	2.661	0.106	0.024
					group	1.387	0.241	0.013
					time	6.934	0.010	0.006
Life	1.05 (0.98)	0.85 (0.92)	0.90 (0.98)	0.57 (0.84)	interaction	0.969	0.327	0.009
					group	1.714	0.193	0.015
					time	14.819	0.000	0.120
Physical	1.39 (1.33)	1.55 (1.51)	1.08 (1.10)	1.35 (1.23)	interaction	0.234	0.630	0.002
					group	1.237	0.269	0.011
					time	3.936	0.050	0.035
Nutrition	0.26 (0.48)	0.35 (0.55)	0.27 (0.45)	0.33 (0.47)	interaction	0.265	0.608	0.002
					group	0.015	0.904	0.000
					time	5.231	0.024	0.046
Oral	0.89 (0.96)	0.97 (0.97)	0.96 (0.87)	0.84 (0.90)	interaction	1.712	0.193	0.015
					group	0.034	0.854	0.000
					time	0.073	0.788	0.001
Outdoor activities	0.69 (0.56)	0.50 (0.62)	0.65 (0.48)	0.49 (0.51)	interaction	0.059	0.809	0.001
					group	0.089	0.765	0.001
					time	8.191	0.005	0.070
Cognitive	0.47 (0.69)	0.55 (0.69)	0.57 (0.74)	0.39 (0.57)	interaction	4.347	0.039	0.038
					group	0.063	0.802	0.001
					time	0.660	0.428	0.006
Depression	0.92 (1.35)	0.84 (1.40)	1.08 (1.35)	0.90 (1.19)	interaction	0.251	0.617	0.002
					group	0.226	0.636	0.002
					time	1.655	0.201	0.015

KCL, Kihon checklist.

**Table 4 ijerph-19-14205-t004:** Comparison of the number of participants who were robust and pre-frail-frail pre-and post-intervention.

		Post					
		Control			Intervention		
		Robust	Pre-Frail-Frail	*p*	Robust	Pre-Frail-Frail	*p*
Pre	Robust	15	5	0.763 ^#^	12	2	0.035 ^#^
	Pre-frail-frail	6	36		9	26	

Pre-frail-frail, pre-frailty and frailty. ^#^ McNemar-Bowker test.

## Data Availability

The database used and analyzed during the present study will be available from the corresponding author upon reasonable request.

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
