# Peer review of "Effect of Home-Based Training with a Daily Calendar on Preventing Frailty in Community-Dwelling Older People during the COVID-19 Pandemic"

_ijerph, 2022, doi:10.3390/ijerph192114205_

Round 1

Reviewer 1 Report

A very interesting and useful article coming from a country that is well known for its centenarian population. Therefore, studies like this one are considered valuable to the scientific community and especially those who work with older adults.

Also, the creation of a daily calendar is considered a clever idea considering the fact that a significant number of older people does not feel comfortable with ICT.

What is the exact number of participants? Authors refer to two different numbers (111 & 126). Please, clarify this.

Reviewer 2 Report

Many thanks for the opportunity to review the paper, as I thought it was fascinating and an excellent contribution to knowledge.  The points below represent only minor areas of context which I would like to see consideration of, and do not represent major criticisms (therefore feel free to argue against them).

They relate to two issues:

1) the focus on the pandemic, and not the related issue of social isolation, as this may add to your relevance beyond the pandemic.

2) the lack of discussion of expected decline in this population, as the fact that certain domain have improved means is as striking as the p-values.

Line 55 - I wonder if there's an opportunity to briefly reference literature here on the impacts of social isolation in older people, as that would reinforce these points.

Line 62 - While I agree with these points, the uncertainty over the impacts of the pandemic might be seen as a potential limitation of the implications for the study.  I think a slight reframing of the importance of these issues beyond this study - see the above point about social isolation - might reinforce the value your study is giving.

Line 68 - guidance exists around the use of language with this population. I would use the term "older people" rather than "elderly people."

Line 79 - was stratification only carried out due to age and gender?  And not existing frailty level or existing health concerns?  This should be justified if so.

Line 202 - I would like to see some of the points in the discussion reframed slightly to consider issues beyond the p-value. Given the population, and the context of the pandemic, it would normally be expected for there to be some form of decline in the participants over three months. I think highlighting domains where this wasn't the case - and the means actually improved - would be quite striking. That the intervention has not just maintained but improved the domains is a key finding for me.

Reviewer 3 Report

The manuscript needs to be improved. Firstly, the authors mention that the frailty factor is influenced by both physical and mental conditions in elderly people. This manuscript does not describe the details; what is the meaning of frailty for the elderly? Why are they issued during a pandemic?; What should be done to fill the gaps compared to the previous studies' methods?

The statistical methods are also unclear to me. This study should clarify the reasons that were used.

Additionally, the authors should compare the results of this study with those of relevant previous studies or reliable findings.

Please find below the specific comments:

Specific comments

Lines 44-45: The introduction section should define what frailty is.

(e.x., physical and mental frailty)

Furthermore, why frailty characteristics adversely affect this population.

Lines 77-79: Even if you did not consider the power analysis to estimate minimum sample size, you may be able to calculate the effect size based on the mean and standard deviation of the previous studies. Describe the recruitment process for your study would be more valuable.

Lines 101-103: If you conducted any intervention programs, the authors should provide detailed information regarding frequency, intensity, and duration. I am not sure how to be conducted each program on a daily basis.

It will be helpful to understand the essential interventions based on the figures that describe each program. It would be great if you could add an appendix.

Table 1: This information should provide more information, as I mentioned earlier.

Line 110: The results of the reliability test for this test should be reported if you are aware of them.

Lines 121-123: Also, please provide a range for the overall KCL score.

Statistical section: Authors should explain why they conducted non-parametric statistical analysis. As an example, the authors might mention that "if the variable did not assume a normal distribution, then we performed a non-parametric analysis to compare the pre- and post-testing results.". This study appears to be a two-way repeated ANOVA design, but why didn't you take this into account?

Also, It would be helpful to explain what the ranges for this effect size mean.

Table 2: The effect size can also be reported in Table 2.

Lines 161-164: Did you consider the comparisons between the intervention and control groups? If so, you should provide a clear description of the statistics.

Discussion section: Here are my suggestions. Initially, I recommend summarizing the main findings of this study, and then discussing the details of the significant findings in the next paragraph. As a final section, the authors may provide suggestions of novelty and creativity, as well as implications and limitations.

Lines 189-191: As I mentioned, the authors should describe the characteristics of frailty, especially what factors are associated with the results of this study.

Lines 193-194: Do you mean that the quality of the acceptability may be low? In this case, the authors should be discussed as a limitation and suggestions should be made as to development strategies.

Lines 199-201: If the authors used the Wilcoxon sign rank test to compare the group differences, this would be an incorrect method. Other statistics, such as independent t-tests and Mann-Whitney U-tests, should be considered as well.

Lines 213-226: I understand what the authors are trying to say, but it should be supported by relevant literature. I am concerned that the authors simply state their opinions based on their own experiences. The study should be compared with similar studies that have been conducted previously. It may be difficult to convince the reader if you are not.

Lines 232-233: It would be helpful for the authors if they provided the KCL score and the status of what they mean.

Lines 234-237: Moreover, the author should explain in more detail how the intervention can improve their physical and mental health.

Lines 247-248: How can the authors emphasize this statement without providing any supporting evidence, such as results concerning acceptability?

Lines 252-254: As I mentioned above, this point may be related to the characteristics of acceptability.

Reviewer 5 Report

This is an interesting work on a possible approach to deterioration associated to the restrictions caused by the COVID-19 pandemic, which are especially suffered by the elderly.

The work is robustly performed.

It is adequately written, although there are some paragraphs that are not well understood. On page 3 (lines 102-105) the authors describe the use of the calendar in the control and intervention groups. That description needs to be revised and improved.

Table 2 is not sufficiently well explained to be understood by itself. The previous frailty situation should be included to improve compression.

The study has some design items that could impact on the conclusions.

In a first place, a before and after study is carried out to assess the impact of incorporating a calendar of activities on the intervention group. However, the epidemiological conditions (and restrictions) are not equal during the pre-intervention period as during the intervention period. Although this fact is commented on by the authors, it supposes a clear and significant limitation of the study that must be exposed and discussed as such.

Second, the characteristics of the study subjects are not sufficiently exposed. To assess the impact of a measure on frailty, it is essential to know more data, that are not showed: at least, sex, pre- and post-intervention nutritional status, socio-family situation of individuals and degree of dependency (e.g., Barthel index). Although some of these data have been commented in results and are discussed, they are not exposed. They should be adequately exposed.

Third, the authors justify that they did not calculate the sample size because the number of individuals could not be predicted before starting the study. In my opinion, this does not justify the lack of sample size calculation, which should be calculated, at least, for the most important studied variable.

Fourth, a 3-month intervention on the evolution of frailty may be too short a period, especially to evaluate the physical component. In my opinion, the authors should discuss the reasons for choosing this intervention period and why they consider it is appropriate to assess the impact of their set of measures on the evolution of frailty in this population.

Round 2

Reviewer 3 Report

The majority of the comments have been accepted in the revised version of the manuscript, and I believe it has been improved.

Reviewer 4 Report

The authors support their main result, prevention of deterioration of cognitive function, using the Kihon checklist that includes only 3 questions for evaluating cognitive function.  The reviewer assesses the method is inadequate and rationale for their conclusion is insufficient to be accepted by IJERPH. 

Reviewer 5 Report

Based on the suggestions made in the previous revision of the manuscript, the authors have improved it significantly.

In this way, paragraphs that were not properly understood have improved their understanding. The authors have assumed the fact that the epidemiological conditions and the restriction measures were not the same during the pre- and post-intervention period, as a limitation of the work. In the same way, they have carried out an adequate calculation of the necessary sample size.

However, the authors do not justify the election of the 3-month period as the duration of the intervention, which was previously solicited; in my opinion, it is an excessively short period to evaluate an action on frailty, especially for physical component of frailty.

Finally, the authors have not included some key aspects of frailty such as nutritional status or degree of disability. In my opinion, the impact of any measure on frailty cannot be properly assessed without adjusting for these factors. Although this aspect is mentioned as a limitation of the work, this limitation lead in that the results cannot be accepted as valid.
